# Scale and Environmental Impacts of Food Loss and Waste in China—A Material Flow Analysis

**DOI:** 10.3390/ijerph20010460

**Published:** 2022-12-27

**Authors:** Li Jia, Jing Zhang, Guanghua Qiao

**Affiliations:** 1College of Economics and Management, Inner Mongolia Agricultural University, Hohhot 010010, China; 2School of Agriculture and Food Sciences, The University of Queensland, Brisbane 4072, Australia

**Keywords:** food loss and waste, material flow analysis, environmental impacts, food balance, food security

## Abstract

Evidence of China’s food loss and waste (FLW) and its increasing impacts on food security and environmental sustainability is urgently needed to guide policy intervention and avoid unnecessary damage to human health and the environment. This paper estimates the scale of China’s FLW in 2019 and assesses the environmental impacts of major food types along the entire food supply chain (FSC) by using a food balance and material flow analysis (MFA) based on existing empirical research. The results show a total FLW of 422.56 Mt which is around 22.37% of total food production (1889.12 Mt). There are also serious environmental impacts on the land footprint (LF), water footprint (WF), and the carbon footprint (CF) estimation (4152.36 × 10^9^ gm^2^, 613.84 × 10^6^ t CO_2_e and 506.07 × 10^9^ m^3^ on average, respectively), most of which are found in foods of animal products, cereals, vegetables and fruit at the stages of consumption, agricultural production, postharvest handling and storage. In addition, the root causes of FLW generation at different levels—micro, meso and macro—were also analyzed. These results will provide significant guidance to researchers and decision-makers on primary data collection and reduction-policy development for China’s FLW.

## 1. Introduction

An increase of about 118 million people are facing hunger in the world [1]. The global agrifood system is facing unprecedented threats and needs to be transformed to achieve sustainable development [2]. At the same time, a total of approximately 1.3 billion tons of food is lost and wasted along the food supply chain (FSC) globally each year [3,4]. The COVID-19 pandemic in recent years has also exacerbated both the hunger issue and waste issue [5], increasing the challenge of achieving Sustainable Development Goals (SDG) targets 2.1 and 12.3. These targets aim to end hunger, achieve food security and improved nutrition, promote sustainable agriculture and halve per capita global food waste at the retail and consumer levels and reduce food losses along the production and supply chains, including post-harvest losses, by 2030 (https://sdgs.un.org/goals (accessed on 13 April 2022). Without considerable policy action or behavioral adjustments, global per capita food waste will double by 2050 and emerging economies will contribute considerably to this issue [6]. A decrease in food loss and waste (FLW) can have a favorable effect on overall output, GDP and employment [7]. FLW incurs severe economic losses and also has a significant negative influence on the environment, aggravating the shortage of arable land and water, and producing astonishing carbon emissions [8,9].

There has been remarkable progress in poverty reduction and social development over the past 35 years in China; however, rates of 9.4 percent for stunting in children represent a significant national burden (https://www.wfp.org/countries/china (accessed on 13 April 2022). At the same time, FLW has been a serious issue in the country [10,11], and is becoming a potential threat to national food security in the context of the changing dietary structure of the population and the increasing risk of international trade in agricultural products. With China already committed to a carbon emission peak before 2030 and to achieve carbon neutrality before 2060, the Central Economic Work Conference has sought faster steps to come up with an action plan, calling for accelerated efforts in cleaner production and sustainable consumption. Due to technology, infrastructure construction, social management and individual consumption perceptions, China’s FLW along the FSC urgently needs to be reduced [12].

FLW is a huge resource and environmental cost [13]. The environmental impacts of the whole life cycle of FLW are serious [14]. One study showed that the waste generated in the food-service sector in Germany is responsible for greenhouse gas emissions of 4.9 million tons of CO_2_ equivalents (CO_2_e), a water use of 103,057 m^3^ and a land demand of 322,838 ha, equating to a total of 278 billion eco-points per year [15]. The ecological footprint of FLW, which imposes an additional burden on regional land resources, especially urban dining waste [10,16], should be a matter of concern. In addition, it is also a huge waste of water resources [17], so reducing FLW can be an effective response to water scarcity [9]. One study estimated that water losses from the wasted consumption of major food items in China account for more than 10% of the country’s total water use [8]. The carbon emissions of FLW are also a threat to a sustainable society. A study found that the carbon emissions of food waste related to six supermarkets in three years in Sweden was 2500 tons [18]. Estimates of carbon emissions from food waste at the national level in Turkey [19] and at the per capita level in the United States [20] have a staggering environmental impact. Taking the United States, as an example, that generates an annual per capita FLW carbon dioxide equivalent of 673(production) + 114(landfill) kg per capita per year, the carbon-reduction effect of changing consumer behavior could be significant.

The environmental impacts embedded in FLW varies for different types of food, with animal products containing a larger environmental footprint and energy. While animal products account for only 34% of the total food waste in the US downstream of the FSC, they generate 60% of the implied energy waste [21]. Similarly, studies in Sweden show that meat waste accounts for only 3.5% of total waste but generates 29% of total carbon emissions [18].

The environmental impact also varies at the different stages along the food supply chain (FSC). For instance, the largest percentage of food wastage occurred at the downstream stage of the FSC in Korea, with about 46% of the total FLW occurring at the consumption stage alone, and 15% of the FLW at the distribution stage. Agricultural production, postharvest handling and storage, processing and packaging stages were responsible for 14%, 11%, and 13% of the total FLW, respectively [22]. It has been found that high income level, the diversity of food consumption patterns, and an increase in industrialization are all strongly correlated with food waste in both developed countries [23] and developing countries [8].

Measuring the flow changes of food and FLW along the FSC at the national level and assessing their resource and environmental costs to find key stop-loss points and intervention programs can provide a scientific basis for alleviating the regional resource pressure of the country [23,24]. There are two kinds of quantification methods used for FLW: direct measurements (approximate and indirect measurements) based on first-hand data, and indirect measurements derived from secondary data [11]. Measurements based on first-hand data can reproduce realistic situations at the micro level [25], thus providing evidence for higher-level estimation, with the main methods being diary, weighing, surveys (questionnaire or interview), waste composition analysis and material flow analysis. Indirect measurement or derivation of food waste from secondary data is another method currently used to estimate the magnitude of regional or national food waste and mainly includes modelling, food balance [17], the use of proxy data, and literature data [26,27]. Most of these studies using secondary data are aimed at estimating the resource–environmental and economic effects of food waste for large regions (countries).

MFA has been used to assess food flow and waste over the kitchen life cycle in consumer-side micro studies [28,29]. However, very few existing studies make use of MFA to account for FLW in China. MFA can used to determine mass and energy flow in the quantification of food efficiency [30,31]. A previous study quantified the food waste per product group along the food supply chain using MFA in the European Union and found that 20.25% of available food was wasted [32]. Comparative studies of research results at the national and regional levels are also very easy to conduct through MFA [32,33]. Data to measure FLW in China are scarce and generally focus on only one segment in the FSC [34,35] or on one type of food [17]. There is a lack of studies that comprehensively assess the FLW status of various food groups along the FSC [36].

The goal of this study is to address this critical data gap through a systematic analysis of large-scale field-based surveys in China based on the literature from recent years. We conducted a material flow analysis to obtain the proportion of losses and wastes of different food types at different stages based on historical data extracted from the literature and the latest data published by Food and Agriculture Organization of the United Nations (FAO). This high-level top-down approach helps enable understanding of the quality flows along the FSC and can be considered as a necessary complement to direct data measurements. In addition, a comparison of the results obtained with existing studies accounting for FLW at a national level was also performed. The environmental impacts were assessed to find the critical stop-loss points and possible intervention options.

## 2. Methods

### 2.1. Definition

This study is based on the five major stages of the FSC, namely agricultural production, postharvest handling and storage, processing, distribution and consumption followed by the FAO [3]. FLW contains two major parts; FL and FW. Since the terms FL and FW are commonly used to describe aspects of food and/or related inedible parts removed from the FSC, users need to decide what constitutes a specific definition of FL or FW for their reporting, depending on their quantification objectives [37]. Therefore, there is confusion over the definition. The FAO has defined that FL and FW occur in different parts of the FSC, with FL occurring mainly in production, processing, storage and distribution, and FW occurring at the end of the FSC [3]. In addition, there are differences in the causes of their generation. FL is due to objective factors such as production technology, whereas the causes of FW are due to subjective consumer factors [38], and we used the definition of both in our study. According to the FAO methodology for calculating FLW, the food removed from the FSC to be valorized as, for example, feed, seed or other non-food is not considered FLW. Therefore, in this study, these flows were accounted for as by-products.

### 2.2. Data Sources and Accounting Methodology

The main statistical data sources used were: (i) the Commodity Balance Sheets (CBS) (FAO statistics) to obtain the production, import, export, stocks, supply and non-food uses of food per product group; (ii) the systematic reviews method to collect, collate, calculate and evaluate the FLW ratio; (iii) the environmental impact factors (carbon, water and land impact factors) taken from the mean values of impact factors obtained using LCA assessment in some research results [39,40,41]. Oilseeds and roots and tubers were classified as having the same environmental impact factors, according to the FAO [42].

#### 2.2.1. MFA Accounting Approach

This work provides a systematized approach to food waste accounting at a national level in China based on MFA, including a detailed compilation of coefficients that can be used to fill data gaps when modelling food waste flows. Moreover, it presents a breakdown of the quantity and environmental impact of FLW per FSC stage of different food groups, enabling the identification of hotspots. China has conducted some informative research studies on various segments of FLW in various food groups in recent years [12,43,44,45] based on large survey samples, which has been important for determining FLW ratios for different food categories in each segment of the FSC when conducting material flow analysis. Most of the studies are on cereals, some others involving roots, tubers, oilseeds, vegetables, fruit, meat and aquatic products, and research about eggs and milk were rare. The accounting path of the study is shown in Figure 1.

#### 2.2.2. Evaluation of the FLW Ratio

This study utilized a systematic review approach to synthesize and evaluate the literature evidence related to quantifying FLW. In a systematic review approach, existing studies were used to explore specific policy-based or practice-based issues in depth [46]. A comprehensive evaluation and comparative analysis of the differences in food-loss rate and waste-rate estimation methods in existing studies was conducted, which, in turn, provided parameter values for the MFA and further estimation of the environmental impacts in this paper.

The China national knowledge infrastructure (CNKI) and Web of Science (WOS) database were used to find the relevant literature. We used {“food loss” or “food waste” or “food loss and waste”} and {“cereals” or “grain” or “staple food” or “roots” or “tubers” or “oilseeds” or “pulses” or “vegetables” or “fruit” or “meat” or “poultry” or “pork” or “beef” or “lamb” or “mutton” or “eggs” or “milk” or “dairy” or “fish” or “seafood” or “aquatic”} and {“agricultural production” or “postharvest handling” or “storage” or “processing” or “distribution” or “consumption” or “food supply chain”} for a topic search of the literature to find academic research papers about China, with the time interval set as 2002 to 6 April, 2022.

The criteria for the literature to be included were: (1) it must be a study on food loss and waste in China; (2) there must be a clear method of data acquisition (research methods, loss-rate calculation methods and estimation results); and (3) a qualitative systematic analysis and evaluation method must have been used to form the results of the systematic evaluation. We primarily used the results of 27 empirical studies shown in Table A1 in Appendix A that contained information on the FLW ratio of different food groups at different stages of the FSC. The measurement of FLW has been a challenge in current research and the literature reflecting FLW ratios in the field is scarce. We extracted data from literature involving large-scale samples based on the main production areas of the corresponding food categories in China. In addition, we hope to make our findings more reliable and to overcome regional, seasonal and measurement errors to some extent by taking the weighted average and median of the findings. The literature that was found for FLW ratios of different food groups along the FSC are shown in Figure 2. For the results of the study of different food groups (e.g., poultry, pork, beef, mutton), we performed a weighted average according to the domestic production and took the median value for the estimated amount of FLW; for the data where it was difficult to obtain the ratio at present, we referred to the FAO estimates for East and Southeast Asia [3]. The food balance of China in 2019 and the calculation of FLW are shown in Table 1.

#### 2.2.3. Assessment of Environmental Impacts Embedded in FLW

The environmental impacts generated from different food types and different FSC segments were assessed, focusing on carbon footprint, water footprint and land footprint. The environmental footprint (carbon footprint, water footprint and land footprint) associated with FLW was quantified based on the footprint factors of each food category and the corresponding total amount of FLW. The footprint factors in this study were derived from other literature which extrapolates the average footprint coefficients of various foods by performing Monte Carlo simulations and linear regression analysis based on the results of a large number of LCA studies on the carbon, water and ecological footprints of various foods selected from the DEEP database [39,40,41]. Oilseeds and roots and tubers were classified as having the same environmental impact factors according to the FAO [42]). The environmental impact factors are shown in Table 2.

## 3. Results

### 3.1. Overall FLW in China

We obtained the flows from the MFA of different agrifood products along the FSC in China during 2019. A total of 1889.12 Mt of food was produced, and the domestic supply quantity was 2028.33 Mt including import/export and change in stock. There was 1340.92 Mt of food for human consumption (adjusted by feed, seed, manufacturing and other non-food use purposes), and only 1012.93 Mt ended up being consumed by humans. The value of the total FLW was 422.56 Mt, that is, approximately 22.37% of China’s total food production does not make it into human stomachs; details are shown in Figure 3. This share is lower than that of Switzerland (34%) [30], Peru (47.76%) [31] and Saudi Arabia (33.1%) [47], but is higher than the EU (20.22%) [33] and Spain (20%) [48]. In the global endeavor for FLW reduction, China can play a big role.

The stages of agricultural production, postharvest handling and storage, and consumption had the largest amount of FLW, with the three accounting for 73.1% of the total FLW (details are shown in Figure 4). From a food group perspective, vegetables and fruit accounted for the highest proportion of the total (56%, 236.66 Mt), followed by cereals (21.3%), roots and tubers (9.7%) and meat (5.6%) (Figure 4b). Vegetables and fruit, and cereals dominated in all the food categories along the all the stages of the FSC. Vegetables and fruit accounted for 56.06%, 72.83%, 25.62%, 77.62%, and 42.09% of the total FLW of all food products in the five stages from production to consumption, respectively. This was especially apparent in post-harvest storage and distribution, mainly due to the perishability of fruit and vegetables, which makes them difficult to store, and they require better packaging and distribution speed for transportation and retail. Cereals were the second largest contributor to FLW, accounting for 22.17% and 35.47% of total in the production and consumption stages, respectively. Roots and tubers also had relatively high losses along the FSC, especially in processing. Although most of the FLW ratios are based on the estimates of Eastern Asia from the FAO as there is a lack of empirical study data), we still need to pay attention to this category due to the large production of roots and tubers (potatoes, primarily) in China. The FLW of meat was mainly reflected in waste at consumption throughout the FSC, which was mainly due to waste from eating out. The FLW amount calculated along the FSC for each food group is shown in Figure 5.

### 3.2. Environmental Impacts of FLW

Details of the environmental impacts of FLW are shown in Figure 6. The average estimate of the national FLW land footprint in China is 4152.36 × 10^9^ gm^2^, with 2939.26 gm^2^ per capita, which corresponds to 276.82 million people’s ecological niches being encroached on, based on 1.5 hm^2^ of ecological footprint per capita in China. Our estimation highlights the unsustainable use of ecological resources in China. An earlier study [41] estimated the LF of major food items for FLW in China as 62.54 million hm^2^ in the consumer segment in China in 2018, which is lower than our value (155.17 million hm^2^ for the consumption of the FSC). This is partly because we studied a more comprehensive range of food, and we used MFA to account for the fact that FLW includes both avoidable and unavoidable food waste. The LF of FLW on consumption and agriculture production are the top two largest contributors, accounting for 37% and 21% of the total along the FSC, respectively. This is mainly due to FW in meat, aquatic products and cereals, and objective factors such as low mechanization ratess resulting in higher losses on the side.

Our estimation showed that the carbon footprint (CF) embedded in FLW in China in 2019 was 613.84 × 10^6^ t CO_2_e, which is equivalent to 6.19% of the total CO_2_ emissions in that year (98.99 × 10^8^ t CO_2_e) in China. China’s FLW-associated CF (613.84 × 10^6^ t CO_2_e) is close to the total national GHG emissions of the United Kingdom, Australia or France [49], highlighting the importance of addressing China’s FLW as part of global climate change mitigation. The three largest amounts of CF related to FLW was in the stages of consumption, agricultural production, postharvest handling and storage, that together accounted for 77.61% of the total carbon emissions. The carbon emissions of FLW of meat, vegetables and cereals were the top three contributors among the food types, and needs intervention to reduce.

For the water footprint (WF), only water use before the food became waste was considered. The total amount of water used to produce the FLW was 506.07 × 10^9^ m^3^, which was similar to the results of an earlier study [50]. A study on the FLW of grains, vegetables and fruit in China in 2010 showed that the total WF of the FLWs was 139 ± 59.7 billion m^3^ [17], which was a little lower than our estimation (233.15 billion m^3^). WF related to the FLW of meat and cereals were two of the highest of all food groups, accounting for 61.7% of the total along the FSC. Considering the implied water footprint of FLW, in addition to the production and consumption stages, the post-harvest and storage stages of vegetables are also priority hotspots for intervention.

Taken together, Figure 6 shows that the greatest environmental impact embedded in FLW is found at the consumption stage for LF, CF and WF (37.37%, 32.74% and 34.97%, respectively), at the agricultural production (21.1%, 22.71% and 22.83%, respectively), and postharvest handling and storage stages (18.1%, 22.16% and 19.68%, respectively). The FLW of meat contributes the most among all food categories to all three environmental footprints, accounting for 33.6%, 37.67% and 35.95%, respectively, of the total LF, CF and WF. The CF of the FLW of vegetables was also relatively high, at 28.32%. The FLW of cereals also had a large environmental impact, accounting for 21.37%, 16.1% and 25.75%, respectively, of the total LF, CF and WF. By understanding the environmental impact of the FLW of different food groups along the FSC, different intervention points can be evaluated to drive towards a more efficient agri-food system.

### 3.3. Comparison with Other Studies

A comparison of the results obtained in this study with other studies about the FLW per FSC stage is presented in Table 3. Figures were expressed in kg of FLW generated per year per capita. The studies selected follow the FUSIONS framework, meaning that both edible and inedible FLW is accounted for and that food used for animal feed was not considered as FLW [51].

The results obtained in this study for the stages of processing and consumption were 45 kg/year/person and 80 kg/year/capita, within the range of values presented in other studies, ranging from 13 to 56 kg/year/capita for the processing stage and 10 to 168 kg/year/capita for the consumption stage. For the agricultural production, postharvest handling and storage, the FLW estimated in this study is higher than those estimated by other scholars for the EU and South Africa [32,52,53], but lower than estimates for Peru [31]. A number of studies have concluded that the FL problem is more severe in developing countries, and that the main occurrence of FLW in developed countries is at the end of the FSC, especially at consumption [42,54,55].

The main difference observed in the comparison is FW at the consumption stage, where the values reported in this study (equivalent to 80 kg/year/person), although lower than in the relevant EU studies [32,52], are significantly higher than in the estimation for South Africa and Peru [31,53]. From another perspective, these differences are regional differences, caused by the multiple impacts of regional economic, environmental and resource factors. As China’s economy grows, food waste is becoming more and more serious at the consumer end of the FSC, especially in the restaurant industry [56,57,58]. The high meat content of FW at banquets and business gatherings has led to a high resource and environmental impact and an urgent need for intervention. It is important to highlight that while other studies also used the MFA methodology, they differ in the determination of the coefficients of their FLW from this paper, which obtained coefficients based on the results of large-scale empirical studies in China in recent years, and that the methodology introduced in this work and the refinement of the coefficients may also lead to differences in estimation.

### 3.4. Uncertainty Analysis

FLW accounting involves multiple sources of uncertainty, such as systematic errors (biases), methodological errors, data processing errors and, in the case of MFA, uncertainties associated with the data used to estimate FLW, which will affect the results [37]. There are uncertainties and limitations in the FAO data [32], and due to the lack of relevant empirical research studies on FL and FW for some food groups in China, some of the coefficients used in this study are not representative of China, but are derived from the average estimates for East Asia from the FAO report.

Most of the FLW coefficients used in this study were obtained from 27 studies retrieved from CNKI and WOS; most of the studies obtained data through interviews and surveys [12,43,44,45,59,60,61,62,63,64,65,66,67,68], and some studies also used modeling approaches based on multiple literature sources to derive FLW coefficients representing a particular food in China [69,70,71]. In order to improve the reliability of the findings of our research and to overcome regional, seasonal and measurement errors to a certain extent, we calculated a weighted average of the results involving different food types and took the median of the literature findings to be the FLW ratio. The overall FLW on cereals was considered to be the most reliable, followed by the proportions of vegetables and fruit, meat and aquatic products. The greatest uncertainty was attributed to the FLW of roots and tubers, oilseeds, eggs and dairy at the processing and consumption stages.

In the MFA analysis, we ignored some of the foods with less volume, such as coffee, tea, pepper, beverages, aquatic plants, etc. Furthermore, since the product of oilseeds is oil, all oilseeds were selected for analysis in the MFA instead of oil, and China’s imports of refined oil products were therefore not considered, possibly underestimating the FW of oil. The FLW from imported products should be analyzed separately in future analyses. The FLW per capita estimated in this paper (299.16 kg/capita/year) is a little higher than the FAO estimate for industrialized Asia (240 kg/capita/year) [3]. Despite the relative robustness of these overall results and the major contribution to the FLW coefficients, further studies are needed to reduce the uncertainty for each stage of the FSC and food group.

## 4. Discussion

### 4.1. Identify Key Points for Intervention

Understanding the FLW status of different food groups at different stages of the FSC and their implied resource–environmental impacts is an important basis for identifying key points for intervention [23,42]. A heatmap of FLW and its environmental impact are show in Figure 7, from which we can understand the relative magnitudes of FLW and the environmental impacts of different food groups along the FSC.

FLW of the food groups in larger amounts should have an intervention at certain stages. This study showed that the amount of FLW from vegetables and cereals was high, especially at the stages of agricultural production, postharvest handling, storage and consumption, and there is a particular need to reduce the amount of waste from the distribution of vegetables. A study based on 1608 households in 28 provinces of China measured the storage losses of cereals in 2015 and found that, amongst the four main crops, corn had the largest storage quantity and the highest storage-loss rate [12]. It was estimated that between 40% and 60% of the total FLW of cereals, vegetables and fruit, which added up to 7.6%, 27.7% and 13.2%, respectively, can still be further mitigated [72].

The results of this study showed that, considering the implied WF of FLW, meat and cereals are the most important categories for reduction, especially in consumption FW. The water footprint of vegetables and fruit is also relatively high. One-quarter of the fresh water consumed for global food production is actually wasted, meaning that unconsumed plant foods represent 174 km^3^ of wasted blue water each year [23]; animal products have a much higher water footprint factor than plant-based products. The meats consumed away from home grow faster than at the home counterparts due to higher income elasticities [73], but FW of meat in the catering industry is much more severe compared to households in China [41], suggesting the importance of reducing meat waste when eating out. Reducing meat waste in hospitality and the food service business may substantially reduce the water footprint of total FLW. Furthermore, the implied water footprint of cereals, vegetables and fruit from production to consumption cannot be ignored.

The LF and CF of FLW also have similar characteristics, with animal products being the category that needs to be focused on. It is worth noting that fish products have a larger ecological footprint and consumption FW, which also implies significant LF. We need to select our interventions according to specific policy objectives to achieve FLW reduction potential using a variety of approaches.

### 4.2. Policy Implication

The complexity and diversity of the various root causes of FLW for specific food groups and specific FSC stages needs to be deciphered, with main factors at three levels globally [4].

The micro-level root causes of FLW are caused by the actions or inactions of individual actors in response (or lack of) to external factors. Meso-level root causes of FLW may occur at the same point in the FSC or may be caused by the way different actors are organized relative to each other, by relationships in the FSC or by the state of the infrastructure. Meso-level root causes can give rise to micro-level root causes or can determine the scope of micro-level root causes. Root causes of FLW at a macro-level explain how FLW is caused by systemic problems, including poorly functioning food systems and the lack of institutional or policy conditions that promote coordination among actors, investment, and the adoption of good operating practices. They also give rise to meso-level and micro-level causes. Ultimately, they are a major cause of FLW on a global scale.

For example, the reasonable selection of varieties that can adapt to local conditions and meet the requirements of the target market is an important issue to be considered in production. Improper selection of varieties can lead to low product quality and can cause a certain degree of production losses, which is a micro-level factor affecting harvest FL. The corresponding industrial problem at the meso-level is the supply and promotion of germplasm resources, and, at the macro level, it is policy factors such as regional and national investment in R&D of breeding and seed selection, and subsidies from relevant departments.

In China, apart from the unpredictable factors related to local resource endowments, agricultural machinery, farmers’ harvesting methods and technology choices are the most important factors affecting the production of agricultural products (cereals, roots and tubers, etc.) relating to FL [61,72]. Insufficient investment in infrastructure and auxiliary equipment can affect the storage and processing FL of cereals. In addition, the lack of a cold chain (to keep food fresh under optimal temperatures) also leads to the loss of fresh agrifood products during the distribution stage, especially distribution FL of vegetables and meat [43,64,72]. An efficient distribution mode for food may reduce the FL of distribution, and the “farm-to-supermarket” mode is better [62]. Furthermore, strict requirements and irrational consumer preferences also promote FL in other stages along the FSC; for instance, irrational consumer preference on “eye-catching, white, and fine” staple food products (e.g., over-polished rice and high-gluten flour) drives manufacturers to implement more processing steps on their agricultural products, which leads to both FL and a loss of micro-nutrients [72]. Consumer preference for fresh products increases the FL of meat and aquatic products and puts higher demands on logistics [43,64].

In our study, there were high quantities of consumer FW and this was embodied by a relatively large LF, CF and WF, which urgently requires intervention. The food waste situation in the Chinese restaurant industry has received much attention from researchers, and some studies have suggested that it largely stems from cultural factors [58]. Compared to other countries, Chinese catering waste is more context-dependent. FW behaviors are influenced by diet, notions of face, and the social culture of extravagance and wastefulness, which leads to a higher amount of waste at events (wedding banquets, etc.) or business gatherings [74]. Comparatively, food waste is low among rural households in China [34,75]. The occurrence of food waste may be related to its opportunity cost and income; if food expenditure is only a small part of total household expenditure, consumers have no incentive to avoid waste [76]. Sound purchase planning [77] (avoiding excessive or impulsive shopping), plate size [28], information interventions [78], psychological factors such as consumer awareness and cognition [79,80] can all influence food waste behavior. To reduce FW, the restaurant industry can improve their food ordering services and meal delivery mode, offering smaller portions and reminding consumers of the quantity, taste and nutritional content of the meals. Normative service standards should be formulated and promulgated at the macro-level for the catering industry. Moreover, education and information interventions on sustainable food consumption and shopping-planning strategies should be strengthened to support consumers to waste less [4].

In summary, we need to prioritize interventions in those food groups and FSC segments that have a greater amount of waste and which contribute to a greater environmental impact. In addition, intervening in consumer FW that is caused by subjective factors is a society-wide value re-orientation that may create long-term and stable effects. Factors associated with FLW in China are shown in Figure 8.

## 5. Conclusions

Despite attempts by international organizations to develop FLW guidelines [37,81] to promote the science, consistency and transparency of quantification (including quantification scope and method), existing studies of China are still lacking, making inter-regional comparisons difficult. This study used food balance sheets for MFA analysis, and determined the FLW proportion coefficients for each stage along the FSC using the results of existing large-scale empirical research on FLW in China. This is conducive to accurately identifying the current situation and environmental impact of FLW, determining key stop-loss points and then determining intervention measures for implementation.

Our study found that the total FLW was 422.56 Mt, that is, approximately 22.37% of the total food production (1889.12 Mt) of China during 2019. This share is lower than those of Switzerland (34%), Peru (47.76%) and Saudi Arabia (33.1%) but is higher than those of the EU (20.22%) and Spain (20%). The amount of FLW per capita in each segment of the FSC is basically in the normal range compared with other countries. For agricultural production, postharvest handling and storage, the FLW estimated was higher than for the EU and South Africa, but lower than the estimation for Peru. The main difference observed in the comparison was the FW at the consumption stage. The estimates of our study (equivalent to 80 kg/year/person) were significantly higher than for South Africa and Peru. These differences may be caused by the multiple influences of regional economic, environmental, and resource factors.

We also assessed the environmental impacts embedded in FLW along the FSC. The average estimate of the LF of FLW in China was 4152.36 × 10^9^ gm^2^, with 2939.26 gm^2^ per capita, which corresponds to 276.82 million people’s ecological niches being encroached on. The CF embedded in FLW was 613.84 × 10^6^ t CO_2_e, which is equivalent to 6.19% of the total CO_2_ emissions of the country (98.99 × 108 t CO_2_e). The total amount of WF of FLW was 506.07 × 109 m^3^. The greatest environmental impact embedded in FLW is found at the stages of consumption, agricultural production, postharvest handling and storage. Animal products, cereals, vegetables and fruit contribute a higher magnitude of environmental impact.

Through FLW quantification using food balance and MFA, this paper provides a preliminary evaluation of FLW generation in China and provides guidance to researchers seeking to collect primary data on FLW or to ensure robust support for decision-making about FLW reduction. The analysis of micro, meso and macro root causes of FLW generation of the key points identified will help to find a path for FLW reduction.

## Figures and Tables

**Figure 1 ijerph-20-00460-f001:**
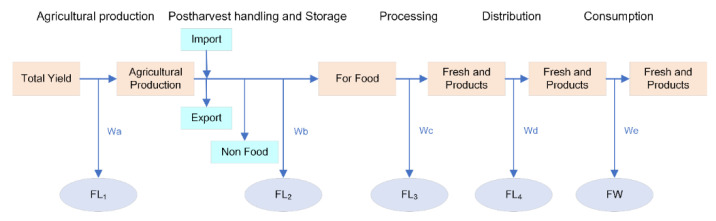
Accounting approach of the MFA.

**Figure 2 ijerph-20-00460-f002:**
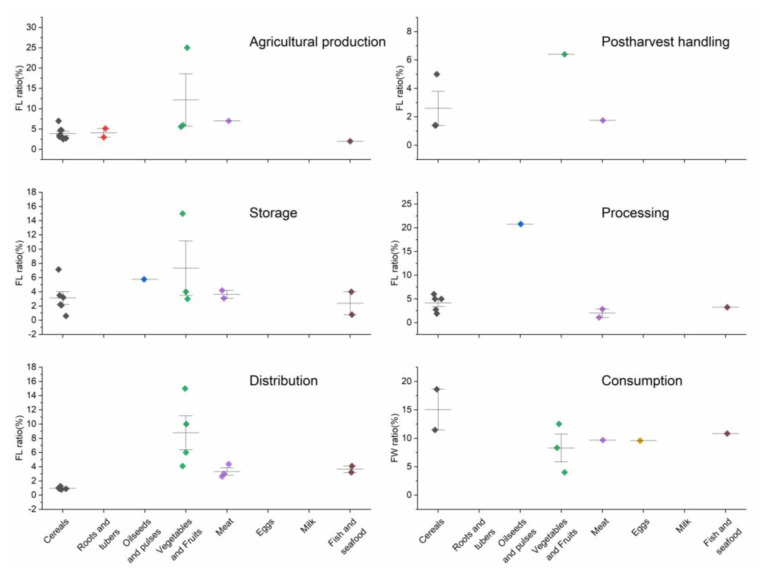
Distribution of literature found for the FLW ratio along the FSC.

**Figure 3 ijerph-20-00460-f003:**
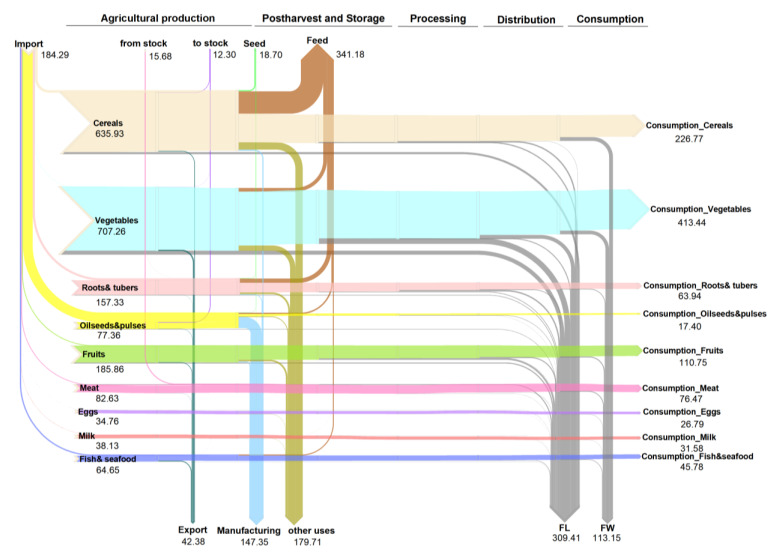
Sankey diagram of the food flows and FLW generated along the FSC. (FL and FW were quantified for each stage of the FSC for nine major agrifood categories (labels on left). Values are in Mt; data are for the year 2019.).

**Figure 4 ijerph-20-00460-f004:**
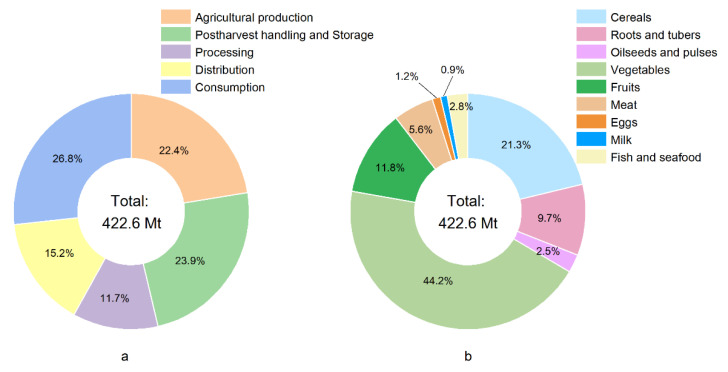
FLW amount share by FSC stages (**a**), and FLW amount share by food categories (**b**).

**Figure 5 ijerph-20-00460-f005:**
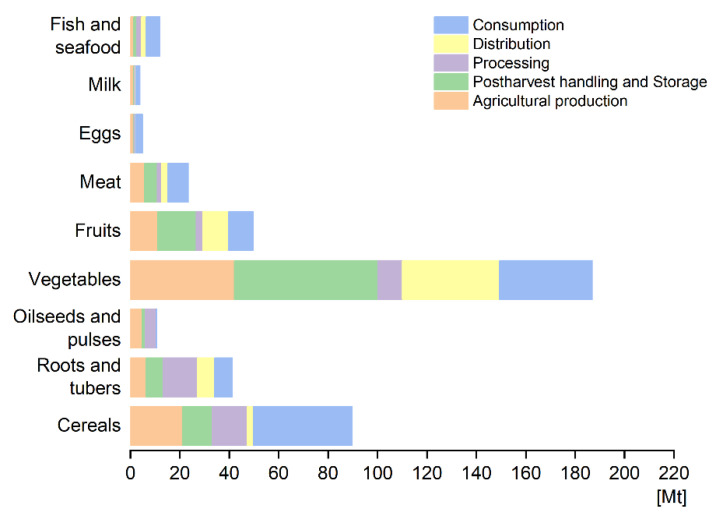
FLW amount calculated along the FSC for each food group.

**Figure 6 ijerph-20-00460-f006:**
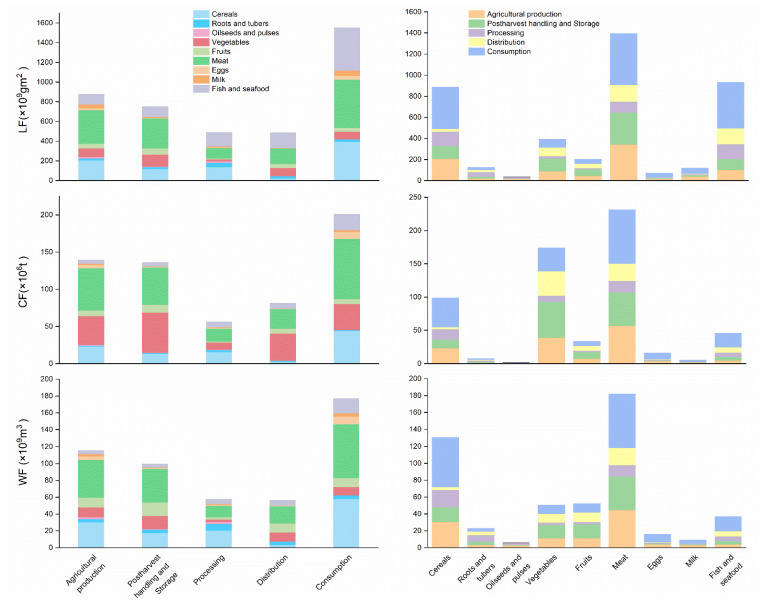
The state of environmental footprints embedded in FLW.

**Figure 7 ijerph-20-00460-f007:**
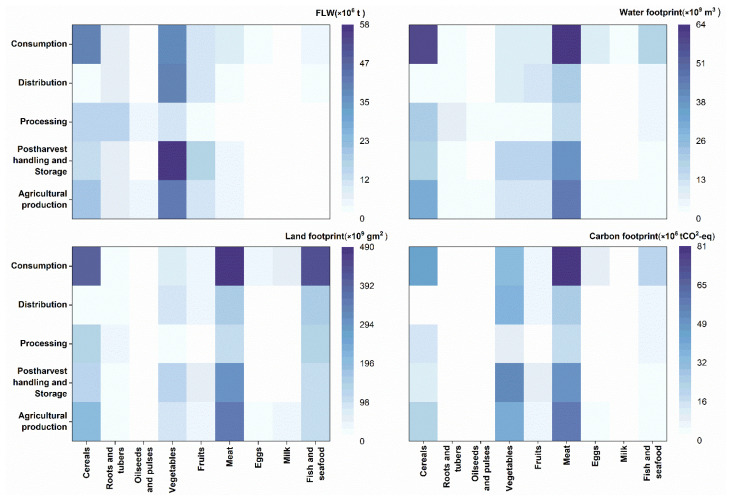
Heatmap of FLW and its environmental impact.

**Figure 8 ijerph-20-00460-f008:**
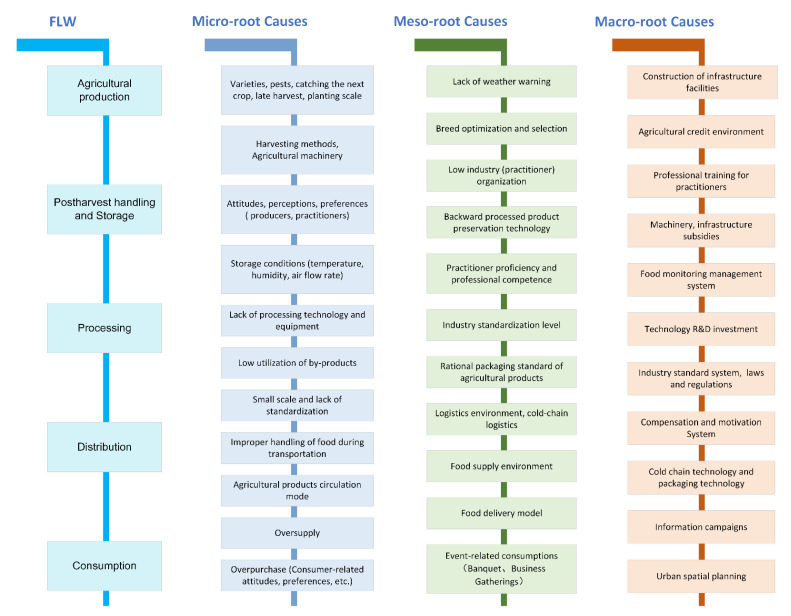
Root causes of FLW in each segment of FSC.

**Table 1 ijerph-20-00460-t001:** Food balance of China in 2019 and the calculation of FLW.

Item	Domestic Production (Mt)	Import (Mt)	Export (Mt)	Change in Stock (Mt)	Domestic Supply Quantity (Mt)	For Feed (Mt)	For Seed (Mt)	For Manufacture (Mt)	For Food (Mt)	Other Uses	FLW (Mt)				
											Agricultural Production	Postharvest Handling and Storage	Processing	Distribution	Consumption
Cereals	614.91	27.49	7.15	6.02	629.23	232.04	12.83	12.61	295.64	76.12	21.02	12.12	14.18	2.42	40.14
Roots and tubers	150.97	22.18	1.10	−1.48	173.53	45.39	2.92	7.62	98.77	18.84	6.36	6.91	13.78	7.03	7.10
Oilseeds and pulses	72.72	100.19	2.40	8.18	162.33	14.80	2.09	114.45	23.31	7.68	4.64	1.34	4.57	0.00	0.01
Vegetables	665.17	2.10	14.76	1.10	651.42	38.29	0.00	0.00	558.30	54.84	42.08	58.06	10.00	39.22	37.57
Fruit	174.80	9.12	6.23	0.02	177.66	0.00	0.00	10.39	149.55	17.73	11.06	15.55	2.68	10.51	10.06
Meat	76.85	7.79	1.32	−10.81	94.13	0.00	0.00	0.00	94.35	−0.22	5.78	5.09	1.76	2.62	8.22
Eggs	33.55	0.23	0.11	0.02	33.65	0.00	0.87	0.00	30.44	2.34	1.22	0.30	0.36	0.15	2.83
Milk	36.79	1.77	0.09	−0.01	38.48	0.18	0.00	2.29	34.16	1.85	1.33	0.34	0.41	0.17	1.66
Fish and seafood	63.36	13.43	9.23	0.34	67.90	10.49	0.00	0.00	56.41	1.00	1.29	1.35	1.77	1.94	5.57
Total	1889.12	184.29	42.38	3.39	2028.33	341.18	18.70	147.35	1340.92	180.17	94.79	101.07	49.51	64.06	113.17

Data source: Food balance sheet from FAO statistics.

**Table 2 ijerph-20-00460-t002:** Environmental impact factors of different food groups.

	Carbon Impact Factors (kg·kg^−1^)	Land Impact Factors (gm^2^·kg^−1^)	Water Impact Factors (m^3^·kg^−1^)
Cereals	1.10	9.87	1.45
Roots and tubers	0.18	3.00	0.56
Oilseeds and pulses	0.18	3.00	0.56
Vegetables	0.93	2.10	0.27
Fruit	0.67	4.05	1.05
Meat	9.85	59.43	7.75
Eggs	3.23	14.41	3.28
Milk	1.43	30.00	2.32
Fish and seafood	3.85	78.25	3.10

Note: Environmental impact factors for cereals are weighted averages of rice, wheat and corn, and environmental impact factors for meat are weighted averages of poultry, beef, lamb and pork, with weights determined by the 2019 domestic production for each of these groups.

**Table 3 ijerph-20-00460-t003:** Comparison of the FLW obtained in this study with studies in other countries.

Source	Year	Country	Agricultural Production (kg/y/capita)	Postharvest Handling and Storage (kg/y/capita)	Processing (kg/y/capita)	Distribution (kg/y/capita)	Consumption (kg/y/capita)	Total Amount (kg/y/capita)
[32]	2011	EU28	64		61	13	119	257
[52]	2011	EU28	26		62	34	168	290
[31]	2007–2017	Peru	109	76	118	56	67	427
[53]	2012	South Africa	57	47	50	39	10	196
This study	2019	China	67	72	35	45	80	299

## Data Availability

The available data was contained within the article.

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
