# Peer review of "Scale and Environmental Impacts of Food Loss and Waste in China—A Material Flow Analysis"

_ijerph, 2022, doi:10.3390/ijerph20010460_

Round 1
Reviewer 1 Report
In the paper titled Scale and the Environmental Impacts of Food Loss and Waste in China: A Material Flow Analysis, the authors provide an interesting perspective on a set of science-based indicators on losses and wastes of different food types at different stages based on the historical data extracted from the literature and the latest data published by FAO. This is a first attempt to estimate on such a detailed and broad scale the FLW of China and assess the environmental impacts of the main FLW types along the entire food supply chain (FSC) using food balance and material flow analysis (MFA) and based on existing empirical research. This paper, after an analysis, ends with the practical application of its findings in the form of several pathways to be considered to address FLW insights for policymakers and other practitioners who want to connect better the governance of FLW with practitioners ` and citizens’ ability to reduce it. You are aware of the limitations of your work. However, this is the first attempt to describe a broad and comprehensive analysis of the determination of key stop-loss points in the estimation and management of FWL in China, followed by a proposal to implement interventions in this area, and from this point of view I consider your work significant.
The article requires some little touch of editing:
- Repetitions
- The quality of Figure 2. Distribution of literature findings on FLW ratio along FSC need a touch of formatting because it is blurry
- Lack of spaces in some sentences, between words and before punctuation marks, e.g., in line 192.
Author Response
Response to reviewer 1:
We appreciate you for your carefulness and conscientiousness. First of all, thank you very much for your recognition of the significance of our article, your suggestions are really valuable and helpful for revising and improving our paper. According to your suggestions, we have made the following revisions to this manuscript:
- Repetitions
Response: Thank you very much for your advice, we have adjusted the full text based on the iThenticate report.
- The quality of Figure 2. Distribution of literature findings on FLW ratio along FSC need a touch of formatting because it is blurry.
Response:Thank you for your advice, we have adjusted the resolution of Figure 2 and other Figures to 600dpi×600dpi. In order to presenting the sources of the findings in this image, we have also added table A1 in Appendix A.
- Lack of spaces in some sentences, between words and before punctuation marks, e.g., in line 192.
Response: Thank you, we have modified it and checked the full text.

Reviewer 2 Report
This manuscript describes food loss and waste (FLW) in China and assesses the environmental impacts using food balance and material flow analysis (MFA). Overall, the manuscript is of interest to readers of this journal. Especially, the finding of this study that around 22.37% of total food production is lost and wasted is impressive. I have a few concerns regarding the present version, which are listed below.
- L173: How much research is reviewed in this study, based on the criteria suggested in the manuscript? I assume there would be regional and seasonal differences between previous studies. Of course, there could be other types of variance to estimate food loss and waste. The explanations on how these differences are compensated in this study should be included in this section.
- L193: LCA is an abbreviation for life cycle assessment. "LCA assessment" is the repetition of "assessment". And LCA is not quite simple. Each LCA has its system set-up and parameters. Merging several LCA results like Table 2 is inappropriate especially when the authors do not provide a description of LCA in previous studies.
- The manuscript has grammatical, spacing, and spelling errors. Authors must fix them before re-submission.

Author Response
Response to reviewer 2:
We appreciate you for your carefulness and conscientiousness. First of all, thank you very much for your recognition of the significance of our article, your suggestions are really valuable and helpful for revising and improving our paper. According to your suggestions, we have made the following revisions to this manuscript:
- L173: How much research is reviewed in this study, based on the criteria suggested in the manuscript? I assume there would be regional and seasonal differences between previous studies. Of course, there could be other types of variance to estimate food loss and waste. The explanations on how these differences are compensated in this study should be included in this section.
Response: Thank you very much for your advice. We primarily used the results of 27 empirical studies. In order to more clearly represent the sources of the findings in figure 2, we have added table A1 in Appendix A(The corresponding citation information in the text is in lines 180-184). Different studies do have certain regional, seasonal, and other measurement errors due to their different subjects, methods, and regions. The measurement of FLW has been a challenge in the current research. Among the quantification studies the literature that reflects the FLW ratio is even rarer. We extracted data from a large-scale research sample based on the main production areas of the corresponding food categories in China. Additionally , we also hope to make the findings of our research more reliable and overcome regional, seasonal and measurement errors to a certain extent by taking the weighted average and median of the findings. This is a compromise choice for us to estimate the FLW along FSC in China. We illustrate in the revised version in lines 184-189 in this section, and explicate this as a limitation in section 3.4 (Uncertainty analysis) in lines 345-348.
- L193: LCA is an abbreviation for life cycle assessment. "LCA assessment" is the repetition of "assessment". And LCA is not quite simple. Each LCA has its system set-up and parameters. Merging several LCA results like Table 2 is inappropriate especially when the authors do not provide a description of LCA in previous studies.
Response: Thank you very much for your advice. Environmental footprints(carbon footprints, water footprints and land footprints)associated with FLW were quantified based on footprint factors of each food category and the corresponding total amount of FLW. We have read a lot of literature in order to find more realistic and objective footprint factors. The footprint factors for this study were derived from other literature. Some of the literature extrapolates the average footprint coefficients of various foods by performing Monte Carlo simulations and linear regression analysis based on the results of a large number of LCA studies on the carbon, water and ecological footprints of various foods selected from the DEEP database. These studies seem to be particularly convincing and their footprint factors were cited by scholars. We selected the coefficients related to our research and applied some weighting to calculate the environmental impacts related to FLW. In summary, we do draw on the results of several studies to get the footprint factors we need,and we have explained this situation in the revised version(lines 205-211).
- The manuscript has grammatical, spacing, and spelling errors. Authors must fix them before re-submission.
Response: Thank you very much for your advice, we have fixed the language and used the English editing services of MDPI to embellish the language.

Round 2
Reviewer 2 Report
I believe the revised manuscript is acceptable for publication.